# The extremely warm summer 2018 in Sweden - set in a historical context

Renate Anna Irma Wilcke[1], Erik Kjellström[1,2], Changgui Lin[1], Daniela Matei[3], Anders Moberg[4], Evangelos Tyrlis[3]

[1]Swedish Meteorological and Hydrological Institute, Norrköping, Sweden
[2]Department of Meteorology and the Bolin Centre for Climate Research, Stockholm University, Stockholm, Sweden
[3]Max Planck Institute for Meteorology, Hamburg, Germany
[4]Department of Physical Geography and the Bolin Centre for Climate Research, Stockholm University, Stockholm, Sweden

*Correspondence to*: Renate A. I. Wilcke (renate.wilcke@smhi.se)

**Abstract.** Two long-lasting high pressure systems in summer 2018 lead to persisting heat-waves over Scandinavia and other parts of Europe and an extended summer period with devastating impacts on agriculture, infrastructure, and human life. We use five climate model ensembles and the unique 263 year long Stockholm temperature time series along with a composite 150 year long time series for the whole of Sweden to set the latest heat-wave in summer 2018 in historical perspective. With 263 years we are able to grasp the pre-industrial time well and see a clear upward trend in temperature itself as well as five heat-wave indicators. With five climate model ensembles providing 20580 simulated summers representing the latest 70 years, we analyse the likelihood of such a heat event and how unusual the 2018 Swedish summer actually was.

We find that conditions as those observed in summer 2018 show up in all climate model ensembles. An exception is the monthly mean temperature for May for which 2018 was warmer than any member in one of the five climate model ensembles. However, even if the ensembles generally hold individual years like 2018, the comparison shows that such conditions are rare. For the indices assessed here, anomalies such as observed in 2018 occur maximally in 5 % of the ensemble members, sometimes even in less than 1 %.

For all indices evaluated, we find that probabilities of a summer like in 2018 have increased from relatively low values in the pre-industrial (1861–1890, one ensemble) as well as the recent past (1951–1980, all five ensembles) to the most recent decades (1989–2018). An implication is that anthropogenic climate change has strongly increased the probability of a warm summer such as the one observed 2018 to occur in Sweden. Despite this, we still find such summers also in the pre-industrial climate, however, with a lower probability.

## 1 Introduction

Long-lasting high-pressure dominated weather resulted in remarkably warm and dry conditions in large parts of northern Europe during summer 2018 (Sinclair et al., 2019). As a consequence, Sweden experienced a very long warm period with an unusually high number of warm days. Similar pressure patterns to those observed in summer 2018 have previously been

shown to be associated with warm temperature anomalies over different parts of Europe (Sousa et al., 2018; Zschenderlein et al., 2019). E.g., Pfahl and Wernli (2012) found that most summer heat-waves (80 %) in Northern Europe and Russia can be associated with atmospheric blocking situations. Sinclair et al. (2019) conclude that the 2018 heat-wave was not intensified

by surface feedbacks and dry soils, instead mainly forced by increased solar radiation due to anomalously clear skies (Räisänen, 2018). The high-pressure situation was established already in May and continued over summer until mid-August interrupted only for short periods mainly in June. According to the SMHI (Swedish Meteorological and Hydrological Institute) weather observations the average temperature over Sweden for the four-month period May–August was on average 2.8 K warmer than the 1981–2010 climatological mean. The sustained period with warm conditions, in connection with little

precipitation, led to a prolonged drought. The hot and dry conditions in summer 2018 in Sweden were associated with severe consequences for people and the environment including: health problems and excessive mortality rate among people (Åström et al., 2019), water shortages with adverse implications for arable land and pastures (Buras et al., 2020) including lack of forage, and unusual large areas affected by forest fires (Krikken et al., 2019). Also, other environmental impacts such as excessive fluxes of carbon dioxide and methane due to extremely warm conditions in shallow parts of the Baltic Sea were

observed (Humborg et al., 2019). Also prior to 2018 heat-waves have been reported to impact Sweden in terms of increased mortality rate among people (e.g., Oudin Åström et al., 2013), ecological consequences (Rasmont and Iserbyt, 2012), and in connection to air pollution episodes (Struzewska and Kaminski, 2008).

As a consequence of global warming, heat-waves have become more frequent and intense in many regions (e.g., SREX, 50   2012; IPCC, 2018; Sippel et al., 2020). In an observational based analysis covering 1950-2017, Perkins-Kirkpatrick and Lewis (2020) find an upward trend in heat wave frequency for large parts of Europe while trends neither in duration nor average intensity across all heat wave days show statistically significant trends. Numerous examples of intense high-impact heat-waves exist in the literature. For European conditions this includes the heat-wave in large parts of southern and central Europe in 2003 and the heat-wave impacting large parts of Russia in 2010 (e.g., Russo et al., 2014). For Scandinavia, in

particular, the vicinity to the relatively cold North Atlantic implies that any shift to a more westerly or north-westerly circulation brings cooler air into the area and efficiently puts an end to any warm period. The large variability of the mid-latitude circulation in this region implies that this is frequently the case and that long extended warm periods in the summer seasons are relatively rare. However, also in this region, heat-waves do occur and are part of climatological conditions. Zschenderlein et al. (2019) report that the maximum number of heat-wave days for their Scandinavian area were 25 for the

1979–2016 period with an annual maximum duration of 12 days for the heat-wave in July–August 1982. Further, they note that the two most intense heat-waves in this area were observed in 1991 and in 2014.

The large year to year variability of heat-waves occurring over northern Europe, implies that it is difficult to assess whether any potential trends or even single extreme events can be attributed to the on-going global warming or if they are a result of 65   the large natural variability. To address such questions large ensembles of climate model simulations have been shown to be

a valuable tool (e.g., Deser et al., 2012, Aalbers et al., 2018, Maher et al., 2019). In two recent studies, Leach et al. (2019) and Yiou et al. (2020) use large ensembles of climate models to address different aspects of heat-wave conditions in summer of 2018 in northern Europe. Determining the probability of an event like 2018 being observed, plus the probability of occurrence conditioned on large scale patterns, Yiou et al. (2020) concluded that the probability of an event such as the one observed in the second half of July 2018 has increased as a result of human-induced climate change. As pointed out by Kirchmeier–Young et al. (2019), and documented also for northern Europe by Leach et al. (2019), the temporal and spatial scales are important for making such an attribution statement.

The aim of the study is to expand findings from previous studies by evaluating a broad range of temperature conditions in Sweden during summer 2018 in relation to the historical climate. We first describe the conditions in 2018 in relation to a number of observations from Sweden for the last 150 years. For one of the stations, Stockholm, the time series extends back until 1756, adding another century to the analysis. Different aspects of heat-wave characteristics; including number of heat-wave events, total number of warm days, total number of consecutive warm days and heat-wave intensity, are assessed. In a next step we investigate the likelihood of such a summer to have occurred in the past century using five large global climate model ensembles, some of which are covering a period since 1860, and others starting in the second half of the 20th century, up to 2018. In this way, we assess to what extent an extreme event like the summer of 2018 may has changed as a result of simulated global warming.

## 2 Methods and Data

The historical context is given by comparing observed conditions in 2018 to observed and simulated climate conditions for: i) a pre-industrial period 1861–1890, ii) a mid 20th century period 1951–1980 and iii) our reference period 1981–2010. For some analysis we also look at the most recent past 30-year period ending 2018 (1989–2018) partly overlapping the reference period. For the longest possible time perspective, we also consider the period 1756–2018 using the Stockholm temperature series. Our analysis focuses on summer months, where the classical summer season, June–August (JJA), is extended with May as Sweden experienced an earlier onset of summer already at the start of May in 2018. This extended summer has also been discussed by Hoy et al. 2020, who called for a summer period from April to September for Europe in 2018. As Sweden spans a wide range of latitudes (55° N to 69° N), and therefore different climates, we analyse conditions north and south of 63° N separately (Figure S1).

## 2.1 Observational data

We use three sources of observational data (Table 1): i) the gridded daily and monthly climatology E-OBS version 19.0e (Haylock et al., 2008) covering Europe at 12.5 km horizontal resolution for the period 1951–2018, ii) monthly averages for Sweden derived from SMHIs observational stations starting 1860 and iii) a long-term record of homogenised daily temperatures derived from the thermometer readings at the Stockholm Observatory starting 1756. Here, we describe the Swedish data sets.

A spatially averaged temperature time series for Sweden has been derived from 35 stations covering the country (Figure S2) with long records, most of which start in 1860 (Alexandersson and Karlström, 2001). For stations with missing data, mostly in the first decades, and for stations where inhomogeneities have been identified (following Alexandersson and Moberg, 1997 and Moberg and Alexandersson, 1997), data from surrounding stations have been used to complement or correct the temperature series. For each station monthly mean anomalies have then been calculated for each month w.r.t. the 1961–1990 average for that month. Next the anomalies are averaged over the 35 stations and added to the 1961–1990 mean for any particular month. In this way we minimise the influence of averaging over different absolute numbers as in the first years of the time series a few of the observational records are not complete. By the use of anomalies we also minimise the influence of changes in location and instrumentation at some of the sites. In the following we denote this as the average temperature for Sweden but we note that it would deviate from any average based on gridded data as the geographical coverage is skewed towards the southern part of the country.

The data from the Stockholm temperature series (Moberg et al., 2002; Moberg, 2020) have been adjusted for i) all years after 1870 to exclude the urban warming trend in the city and for other inhomogeneities detected in homogeneity tests against surrounding reference station data and ii) before 1859 to adjust for a supposed warm bias of the observed temperatures during summer. The adjustment for the urban warming trend and other inhomogeneities, is applied such that the most recent homogenised temperatures are made somewhat colder than the non-homogenised temperatures in order to make the entire series representative of the rural conditions that prevailed before the mid-19th century. The size of the adjustment changes with time and varies over the year. It causes homogenised temperature data after 1967 being on average 0.8 K colder than non-homogenised temperatures for annual-mean data and on average 1.0 K colder for May–August. The second adjustment term is the same for every year before 1859, where it increases from 0 K at the start of May to -0.7 K in June and July and then decreases back to 0 K again at the end of August. This warm bias is likely due to poor protection of thermometers against radiation during May–August in the period before 1859 (Moberg et al., 2003). We use data with the same adjustment as in Moberg et al. (2005). This adjustment does alter our results only marginally and does not affect the conclusions we draw.

### 2.2 Climate model data

We use five GCM ensembles as listed in Table 2: i) one member (r1) from each GCM in the CMIP5 (fifth phase of the Coupled Model Intercomparison Project) multi-member ensemble (Taylor et al., 2012), ii) the 100-member MPI–ESM grand ensemble (Maher et al., 2019), iii) the 50-member CanSISE grand ensemble (Kushner et al., 2018; Scinocca et al., 2016), iv) the 16-member EC-EARTH (CMIP5 version) ensemble from KNMI (Aalbers et al., 2018), and v) the 50-member EC-EARTH (S-LENS (Doescher et al. in prep); CMIP6 version (Eyring et al., 2016)) ensemble from SMHI. The climate models have been run with forcing conditions representing those observed in 1860–2005 following CMIP5 and CMIP6 protocols respectively. From 2006 onward the climate models (i–iv) have been forced by the RCP (representative concentration pathways) scenarios representing future conditions (RCP, Moss et al., 2010). For the S-LENS ensemble, EC-Earth has been forced by the SSP5-8.5 scenario from 2016 and by observed forcing until 2015 (Riahi et al., 2017). Though SSP5-8.5 is not identical to RCP8.5 forcing, it is the trajectory which is closest to the RCP8.5 pathway (Meinshausen et al., 2020). Most models have been forced by the RCP8.5 (or SSP5-8.5) scenario while the MPI-GE has been forced by RCP4.5. For all models, but EC-EARTH S-LENS, the scenario forcing starts with year 2006. As differences in radiative forcing between these scenarios are small in the first decades of the 21st century, we neglect any differences between the scenarios. Details related to time periods, temporal resolution of temperature data, forcing conditions and size of the respective model ensembles are given in Table 2. In total, we analyse 294 simulations expanding the sample size for each 30 year period to 8820 (30 summers times 294 simulations). The larger sample increases the possibility of assessing statistically robust probabilities of heat events.

### 2.3 Variables and indices

All analysis is based on either daily or monthly temperature data. We look at both daily average and daily maximum temperatures, as well as monthly means of daily average temperature. To assess the average temperature for the summer season we use monthly mean temperatures for four individual summer months (May, June, July, and August) separately and for two summer seasons (JJA, MJJA; seasonal averaged). Temperature anomalies are calculated against the reference period 1981–2010.

Furthermore, three warm day indices, based on daily values, are used to assess the temperature variability during summer: i) the "total number of warm days per year" (totWarmD), ii) the "maximum number of consecutive warm days per year" (max_conWarmD), iii) the "number of heat events" (tot_event), where an event is defined as minimum 3 consecutive days of $T_{max} > threshold$.

For simplicity reasons we define a warm day as a day $i$ with $T_{max,i}$ greater than a relative threshold. The threshold is the 95th percentile calculated of all $T_{max}$ in May to August (MJJA) during the reference period (see Eq. (1)).

$$T_{max,i} > p95(T_{max,(MJJA\ 1981-2010)}) \tag{1}$$

This simple definition based on a relative measure is chosen to make it possible to compare conditions in different parts of Sweden to each other. For example, a perceived heat-wave in the colder north of Sweden, may not even appear in an analysis involving an absolute threshold representative of conditions in southern parts of the country like, e.g., summer days defined as days with $T_{max} > 25°\,C$. Other examples of benefits with a relative measure involve comparison of coastal and inland

stations or between low and high altitude stations for similar reasons.

Additionally we calculate two heat-wave indices commonly used. The Warm Spell Duration Index (WSDI, Orlowsky and Seneviratne, 2012) that can be compared to max_conWarmD, that differ in their definition of the warm day. The WSDI is calculated based on an individual threshold for each day of the year (doy). A warm day is defined as a day with $T_{max,i}$ larger

than the 90th percentile of $A_i$, as defined in Eq. 2 (from Eq. 1 in Russo et al., 2014).

$$A_i = \bigcup_{y=1981}^{2010} \bigcup_{d=i-15}^{i+15} T_{max,y,d} \tag{2}$$

Where $\cup$ denotes the union of sets and $T_{max,y,d}$ is the daily maximum temperature of the day d in the year y.

The WSDI is determined as the maximum number of consecutive warm days (larger than 3), i.e. for a given year (or season), the WSDI is the longest duration of any such heat-wave event.

The second heat-wave index is the Heat Wave Magnitude Index (HWMI, Russo et al., 2014), which uses the same warm day definition as WSDI. Whereas the WSDI takes into account only the duration, the HWMI also considers the magnitude of the heat-wave. The HWMI takes into account multiple local maxima of an event by summing them up and mapping them to a probability (called magnitude) related to annual maximum magnitudes of the reference period. By that it weights the duration more than the absolute maximum temperature of an event. This relates to the heat stress which builds up with many

warm days in a row rather than a couple of very warm days in a row (e.g., Notely et al., 2018). A more detailed description of how to calculate the HWMI can be found in Russo et al. (2014).

The analysis is carried out for all model ensembles where respective data is available. I.e. the pdf analysis based on monthly data includes all five ensembles (c.f. Table 2 and Table S1), whereas the warm day indices are based on daily temperature values which are not available from MPI-GE.

## 3 Results and discussion

The analysis is performed in four steps. First the observed temperature climate of summer 2018 in Sweden is set into a long-term 250-year perspective by comparing to the Stockholm series and in a wider Swedish context by comparing to observation data from SMHI and to E-OBS. Next, we investigate to which extent the five GCMs represent warm summer

conditions over the reference period (1981-2010) by comparing the GCM ensembles with the observed climate. We also investigate to what extent conditions in Sweden in the summer of 2018 fall within the simulated ensembles and thereby infer

to what extent it could be considered extreme. Finally we look at the GCM ensembles to address the question as whether a warm summer, such as that observed in 2018, would have been probable in a historical context by comparing to pre-industrial (1861–1890) and mid 20th-century (1951–1980) conditions. The analysis is first presented for monthly and seasonal mean data in 3.1 and then repeated in 3.2 for the indices based on daily temperature data.

### 3.1 How extreme was the summer of 2018 in Sweden on a monthly and seasonal basis?

For Stockholm, the mean MJJA temperature of 2018 was 17.8° C, which is 3.0 K above the mean for 1998–2017 or 4.0 K above the 1756–1900 mean. The temperature anomalies in Stockholm for each day in 2018 w.r.t. to the long-term 250-year climatological average of 1756–2005 are displayed in Fig. 1. The figure clearly shows that the extended summer, ranging from May to mid August, yielded a large number of days with temperatures by far exceeding the long-term mean. As shown by Sinclair et al. (2019) the large-scale circulation was dominated by persistent blocking high-pressure systems over Scandinavia in May and large parts of July and early August. In June, the situation was different with more low-pressure dominated weather in northern Scandinavia while the southernmost parts were under the influence of a high-pressure ridge over Western Europe. This resulted in lower temperatures, mainly in northern Sweden, but also episodically in southern Sweden associated with intrusions of cooler air from the west and northwest. The episodes of relatively colder conditions in June can be seen in the Stockholm record as illustrated in Fig. 1 by blue downward facing spikes.

For the full four-month period MJJA we note that more than 35 % of the days were above the long-term (1756–2005) climatological 95th percentile calculated for each day (upper grey line in the figure). This was indeed a unique feature for 2018. Only one additional year, the year 2002, exceeds the 95th percentile with 20 % of the days in a full May–August season. Additionally, only 10 other years exceed the 95th percentile with 15 % of their days for MJJA, indicating that a summer, such as 2018, with a very large number of warm days, is rare in a long-term historical perspective. This result is confirmed by the study of Hoy et al. (2020) analysing 67 long time series all over Europe, where 33 stations show the summer half year of 2018 to be the warmest on record. Hoy et al. (2020; Fig. 5) also show that Stockholm time series peaks at 2018 for 9 out of 10 of their analysed heat-wave indicators (Fig. 5 and 10 in Hoy et al., 2020). An example is hot days (HD) with maximum temperature above 30° C which was observed with 8 days over the previous record (18 days) as well as the heat-wave duration with 22 days compared to 11 days previous record and a heat-wave intensity ($HW_{95}$, sum of daily excess maximum temperatures >P95) of 65 K compared to 35 K in 1975 (Hoy et al., 2020, Fig. 7). Another peculiar observation in Figure 1 is that most individual daily mean temperatures in summer 2018 were not exceptionally warm when viewed one-by-one. In the May–August season, it is rather the large total number of days with temperatures above the long-term climatological 95th percentile that was unique, of which most occurred in May and the second half of July.

Zooming out to the larger scale by exploring the observed monthly data over Sweden, we note that the long-term MJJA 2018
average was 2.8 K above the 1981–2010 mean. This is more than 0.7 K above the second warmest MJJA, recorded in 2002.
Compared to the historical data from 1860, there is a clear indication of higher average seasonal mean temperatures in the
last decades. The mid 20th century period 1951–1980 is 0.6 K warmer on average than the preindustrial 1861–1890 mean.
The difference compared to 1861–1890 increases to 1.2 K for 1981–2010 and to 1.5 K for 1989–2018. Furthermore, 12 out
of the 16 (corresponding to the 10th percentile) warmest MJJA periods in the last 160 years (1860–2019) occur after 1988.

Probability distributions (pdfs) of the four single summer months (May, June, July, and August) from E-OBS temperature
data in Fig. 2 (a–d) further illustrate the rarity of the weather situation in 2018 in southern Sweden. This can be seen as May
(Fig. 2a) and July (Fig. 2c) 2018, indicated by the dotted line, sets the 100 % mark on the extreme warm tail of the
observation distribution for the last decades (grey histogram). June (Fig. 2b) and August (Fig. 2d) were also warmer than
normal in southern Sweden, but not within the extreme tail, with June just above the 95th percentile and August well below
the 90th percentile. For northern Sweden the picture is similar for May, July, and August, but here June 2018 was in fact
colder than on average, by about 1 K (see Fig. 3), due to low-pressure intrusions. We note that the monthly mean
temperatures of May 2018 are the highest observed so far for May in all 25 Swedish provinces according to SMHI's
observations (not shown).

Pooling the single months together to JJA and MJJA distributions (Fig. 2 e–f), 2018 is the most extreme summer in southern
Sweden in the past 30 years according to the E-OBS data set. Also in northern Sweden, summer 2018 was more than 1 K
warmer than on average but far from being among the warmest years (Fig. 3 e–f). The extended summer period (MJJA) 2018
is just above the 95th percentile in northern Sweden, as the second warmest event in the period 1989–2018.

The climate simulations shown in Figs. 2 and 3 reveal a smoother picture compared to the observations as we have pooled
together all members of one ensemble in one probability distribution (solid coloured lines).

For the recent past period (1989–2018) five GCM ensembles (see Table 2) were available and despite their varying ensemble
size and different models, the probability distributions peak at similar monthly and seasonal mean temperature anomalies.
We note, however, that even if the average anomalies are similar there are differences in the spread of the individual
ensemble members, as reflected by different width and height of the probability distributions.
The 1989–2018 ensemble medians show temperature increases exceeding those of the reference period (1981–2010) by
between 0.19 K in E-OBS to 0.44 K in EC-Earth-LENS for MJJA conditions.

Compared with the spread in observed temperatures as derived from E-OBS, all five GCM ensembles include summers as
warm (or as cold) as any individual observed summer during the 1989–2018 period, both for individual months and for the

three and four month long summer seasons. May 2018 is, however, an exception to this as it is warmer than all of the 100 simulated May months in the MPI-GE for southern Sweden. The fact that the GCM ensembles generally show a larger spread than that of the observations is not surprising as they represent a large number of realisations, and thereby many 30-year periods representing the recent past climate while the E-OBS data only represents one such period. The assessment presented here is not a full evaluation of the ability of the models to represent summertime temperatures in Sweden. However, the coverage of the E-OBS probability distribution within the GCM ensembles lends confidence in the climate models.

For northern Sweden, the models show a similar agreement as in southern Sweden regarding the position of the peak for each month/season (Fig. 3). Even though the observed May and July temperatures are the highest in the last 30 years, the model ensembles indicate that temperatures as high as those observed in 2018 are more likely than in the South. Probabilities are, however, still low, being below 5 % for most of the model ensembles. Compared to southern Sweden, June and August 2018 were relatively mild in northern Sweden. For June, 2018 was even on the cooler side of the monthly temperature distributions.

Even if only May 2018 is completely outside of one of the GCM ensembles in southern Sweden, it is clear that both May and July, as well as the JJA and MJJA summer seasons in 2018 stand out as being warm or very warm also in the context of the large model ensembles.

Three ensembles include pre-industrial data, giving us the opportunity to compare the recent past climate and the year 2018 with the pre-industrial period 1861–1890. We also compare with the mid 20th-century period 1951–1980 to be consistent with the analysis of daily data in section 3.2 below. Here, the observed May 2018 temperatures in southern Sweden are outside of all climate model ensembles for 1861–1890 apart from the CMIP5 multi-model ensemble (dashed lines and bars in Fig. 2). However, the probability of such a warm month is below 1 % also in that ensemble (Fig. 2a). The other two GCMs, MPI-GE and EC-EARTH, do not include such a warm May in any of their ensemble members, not even for the mid 20th-century period, 1951–1980 (not shown). Again, pooling together the months to MJJA actually none of the ensembles simulate temperature anomalies as high as in 2018 for southern Sweden neither in the pre-industrial period (Fig. 2f dashed lines) nor in the mid 20th-century. This also indicates that CMIP5 temperatures like in May 2018 fall only into the distribution because of the chosen fit of the Gaussian pdf estimate. (The pdf curve continues smoothly to zero even if there is no value at the end of the tail.) Even if conditions in northern Sweden were not as extreme, with the relatively cooler June and August, it is clear that the probability of a summer (MJJA) as warm as that observed in 2018 is less than 1 % in all ensembles (Fig. 3).

**3.2 How extreme was the warm summer of 2018 in Sweden on an event-basis?**

As a heat-wave is not only perceived by monthly averages, we calculated five different indices to account for different characteristics of extreme and long-lasting events (cf. section 2.3): the total number of warm days (totWarmD), the maximum number of consecutive warm days representing the longest heat event (max_conWarmD), the number of heat events lasting at least three days (tot_event), the Heat Wave Magnitude Index (HWMI) and the Warm Spell Duration Index (WSDI). These indices are calculated for the extended summer period, MJJA. As they are all based on maximum daily temperature data, which is only available from 1951 from most of the simulations (see Table 2), we here compare the mid 20th-century (1951–1980) to the present day period (1989–2018).

Figure 4a reveals that a total of 28 warm days (totWarmD) are found for southern Sweden in MJJA 2018 according to E-OBS. It stands clear that this puts MJJA 2018 on the extreme end and well above any other years. Comparing the observations for 1989–2018 with 1951–1980 shows that the distribution of totWarmD has widened at the same time as the mean and median (red mark) has been shifted towards more warm days. Also the model ensembles agree on that message, with CanSISE showing the strongest change signal between the two periods. Looking at the most extreme summers it is clear that all models indicate an increase as seen both by a shift of the 95th percentile (upper whisker), and by larger absolute maxima. For three out of four model ensembles more summers with more than 28 warm days can be found in the present day period compared to the mid 20th-century period. We also note that both observations and models indicate that there is a tendency for summers without any warm days to become less frequent with time. From Fig. 4b we see that 28 warm days would only have been reached once in the simulated pre-industrial climate, in the 1860s, though only represented by one ensemble.

The maximum number of consecutive warm days (max_conWarmD) reflects another aspect of warm days. This is important as extended periods with warm conditions have a stronger impact on human health, and also on other parts of nature, than a single heat day (e.g. Oudin Åström et al., 2013; Rasmont and Iserbyt, 2012). Figure 4d shows the time series for max_conWarmD where the thin coloured lines represent the single ensemble members, the bold lines the ensemble means, and the horizontal dotted black line indicates the number for 2018 as calculated from E-OBS. The figure also shows that observed variability from E-OBS (black solid line) lies within the variability of all simulations and that 2018 does not stand out as being among the most extreme summers in the period. The sharp peaks for the single ensemble members indicate high year to year variability of max_conWarmD for each member, as for the other indices too (right panels in Fig. 4).

The warming signal starting at the last decades of the 20th century is clearly visible for all four ensembles, but in particular when referring to the long time series of EC-EARTH. The time series also show that summer 2018 max_conWarmD value is passed much more often in recent years than in earlier times, shifting the summer 2018 event towards the centre of the

distributions. Figure 4c shows the change of all ensembles towards longer events with more consecutive warm days going from 1951–1980 to 1989–2018. However, it is clear that summer 2018 is still outside the whiskers in the figure indicating that it is above the 95 % range for all ensembles. Again, like for the totWarmD (Fig. 3a), the lower whisker changes towards the present day climate by starting to lift from zero (lower boundary).

The median max_conWarmD is very similar for all ensembles at about 2 days and 3 days, respectively for the 1951–1980 and 1989–2018 period (cf. black median lines in the boxes in Fig. 4c). The maximum value ranges from 12 days to 20 days for all ensembles and both periods. These results are very similar to Zschenderlein et al. 2019 who found a maximum heat-wave duration of 12 days. We find this close similarity despite differences in data set, periods, exact definition of index and regions not being identical.

Also the total number of heat events (tot_event) reveals that summer 2018 was an unusual event with more heat events than covered by ensemble simulations for the historical time period (Fig. 4e and f). It is also leading the extreme tail even for the present day climate. The ensemble means show a clear upward trend at the end of the 20th century and the pooled distribution indicates a similar detachment from zero as discussed above for max_conWarmD. Also the observations show a detachment from the zero lower limit in the most recent 20 years (Figure 4f). As for the tot_WarmD (Fig. 4b) only one year in pre-industrial climate has been simulated above the summer 2018 value by the EC-Earth model (Fig. 4f).

The picture of more extensive and intense heat-waves as indicated above is manifested in all indices. Also, the more complex heat indices WSDI and HWMI support this finding. Figure 5a shows the WSDI for each ensemble pooled in box-whiskers and the WSDI for 2018 based on EOBS as a dotted horizontal line. Also for WSDI the summer 2018 is on the extreme tail for all ensembles and periods. It again shows a clear upward shift of all distributions towards the summer 2018 value. The time series analysis of EOBS indicates that 2018 stands out relative to any other year observed since 1951. At the same time, however, WSDI numbers as high as that observed in 2018 do show up in the model ensembles for 1989–2018, reaching almost the 5 % level in the CanSISE model.

Also HWMI (Fig. 5c and d) follows the main picture described above. The summer 2018 value of 2.5 indicates a moderate heat-wave magnitude (Russo et al., 2014) that does occur in all ensembles but only with relatively low probabilities. Figure 5d shows that observed HWMI is varying between 0 and 2.5 for southern Sweden with a mean increasing from about 0.7 in the mid 20th century to 0.9 in the beginning of the current century.

**3.3 Discussion**

In this study we assess to what degree summer 2018 in Sweden can be considered as being an extreme event. Without doubt the summer was remarkably warm and in some aspects way beyond what has been observed, for Sweden as a whole for the last 160 years and for Stockholm in a 250 year perspective. However, the results also clearly show that the degree of rarity of the summer is very different depending on which measure is investigated. The monthly mean temperature for May and the number of warm days above the 95th percentile in MJJA are examples for which 2018 is the most extreme summer observed so far. For other indices, like the WSDI, results indicate less extreme conditions compared to other years. Such differences may simply be the result of looking at different aspects of warm summers but may also be a result of different methods, like in this case differences between using a fixed threshold relative to one index based on a day-to-day evolving threshold. As an example of how results may be different depending on definition, we note that in case of not counting May as a summer month, focusing on the more commonly used generic three-month JJA summer, summer 2018 was not as remarkable. In particular, in light of the large ensembles such a particular warm JJA summer is found in single members of all assessed climate model ensembles even in the pre-industrial climate. Such a definition would make the study more comparable to other summer studies, but missing the impact of long-lasting heat events, when not including an early onset of summer temperatures.

An implication of our findings is that attribution statements based on specific features of a part of a season can not be taken as being representative in a more general perspective of the full season and vice versa. Yiou et al. (2020) could demonstrate the thermodynamic contribution of human-induced climate change for the Scandinavian heat-wave 2018. They found a +5.4 K temperature anomaly for the second and warmer part of the heat-wave (19 days at the end of July 2018) over Scandinavia (including Norway, Sweden, Denmark, and Finland) relative to the reference period of 1981–2010. In particular, they found each single day of this period was at least 3 K above the climatological mean. Yiou et al. (2020) also made a probability analysis based on two GCM ensembles and two RCM ensembles and found a heat event like those 19 days 5 to 2000 times more likely in present day climate than in the pre-industrial period. This agrees well with our analysis of the five large GCM ensembles.

Leach et al. (2019) state that the 2018 heat-wave over Europe would not have happened without human-induced climate change. For Europe they found an event like 2018 summer to be 10 to 100 times more likely in present day than in pre-industrial climate.

For all aspects of summer temperatures assessed in this study the five model ensembles present a relatively coherent picture with central values (medians) and ensemble spread (e.g., standard deviation or interquartile ranges) being similar. However, for the more extreme values like the 95th or 99th percentile, differences between ensembles tend to be larger with some more systematic differences between them. For instance, the CanSISE and the EC-Earth-LENS ensemble tend to give

significantly stronger warm extremes than the other ensembles from the previous version of the EC-Earth model and the MPI-GE. We note that for all model ensembles assessed here, there are, with few exceptions, always individual summers as extreme as 2018 in the 1989–2018 period. It is difficult to judge, based on the relatively simple evaluation presented here, whether the GCMs are performing well implying that such summers are, even if rare events, still an existing type of summer in today's climate, or, if the models are overestimating the chance of such a summer. This fundamental uncertainty restricts us from making a formal attribution analysis as we simply don't know which climate to evaluate the models against.

All models suffer from simulation biases. For summer temperatures all models show a certain cold bias over northern Europe which range from -0.2 K to -1 K (not shown). However, as the analysis is done for each ensemble separately and focussed on anomalies, the impact of any systematic bias is reduced.

It is known that GCMs have difficulties to capture many aspects of blocking (e.g., Davini and D'Andrea, 2016 ; Dawson and Palmer, 2016). Studies on atmospheric blocking representation in the GCMs used here, agree on an underestimation of blocking frequency mostly in winter, but also in summer (EC-EARTH: Hartung et al., 2017; CMIP5: Woolings et al., 2018, Masato et al., 2013, Dunn-Sigouin and Son, 2013; MPI-GE: Maher et al., 2019, Müller et al., 2017; CanESM: Schaller et al., 2018, Brunner et al., 2017).

Increasing horizontal and vertical resolution of GCMs can strongly improve the ability to capture atmospheric blockings (Hartung et al., 2017, Jung et al., 2012). However a positive effect of increased resolution could not be confirmed for summer blockings (Schiemann et al., 2017). Sousa et al. (2018) found that the frequency of summer blocking (23 % of the days, and only 9 % over Scandinavia) is less than of winter blocking (35 % of the days). The small sample size of summer blockings over northern Europe handicap statistical analysis, hence the amount of references on that topic is low.

For this study we focus on the advantage of large ensembles that increase the sample size for statistical evaluation of high temperature events.

A more extensive model evaluation exercise including a rigorous test of the model's ability to simulate extended periods with high-pressure situations such as that observed in 2018 including the temperature conditions in such episodes is planned for a forthcoming study.

**4 Summary and Conclusions**

We analyse temperature conditions in Sweden during summer 2018 in comparison with the observed climate. The results clearly show that summer 2018 stands out as an unusually warm event in relation to the observed climate. We have also shown that the degree of rarity depends on which measure is evaluated. For instance, we note that the average temperature of the month of May was the most exceptional compared to observations. Also average July temperatures as well as the four-month average May–August temperatures are far above previous maxima for southern Sweden while June and August temperatures were more modest. Some other indices, like the warm spell duration index (WSDI) and heat-wave magnitude index (HWMI) show less extreme conditions.

We also compare the summer 2018 with a large number of climate model simulations. Five different global climate model ensembles, with in total 294 climate model simulations have been assessed. This gives us the opportunity to set the observed summer in 2018 in perspective of 8820 summers over the period of 1989–2018. We find that conditions like those observed in summer 2018 do show up in all climate model ensembles. An exception is the monthly mean temperature for May for

which 2018 was warmer than any member in one of the climate model ensembles. However, even if the ensembles hold individual years like 2018, the comparison shows that such conditions are rare. For the indices assessed here, anomalies as those observed in 2018 occur less than once in 20 in all ensembles except for the total number of warm events in one of the ensembles. For some indices, the anomalies are even more rare and do not occur more frequently than once in 100. We note that there are very large differences between the different models in what degree of probability a summer like 2018 would

have. As a consequence, we do not assign specific probabilities but instead discuss the results more qualitatively.

As the simulations also cover historical periods further back in time we can assess to what extent conditions have changed over time as a consequence of increased greenhouse warming. For all indices evaluated we find that probabilities of a summer like that 2018 have increased from relatively low values 1861–90 and 1951–80 to the most recent decades (1989–

435 2018). An implication is that anthropogenic climate change has strongly increased the probability of a warm summer such as the one observed 2018 to occur in Sweden. Despite this, we still find such summers also to occur in models in the pre-industrial climate, although with a lower probability.

**Data availability**

MPI Grand Ensemble is now available openly:

https://esgf-data.dkrz.de/projects/mpi-ge/

CanSISE data from CanESM2 model is available online via:

http://crd-data-donnees-rdc.ec.gc.ca/CCCMA/products/CanSISE/output/CCCma/

EC-EARTH ensemble from KNMI is available:

https://climexp.knmi.nl/KNMI14Data/CMIP5/output/KNMI/ECEARTH23/rcp85/

EC-EARTH-LENS is produced by SMHI and openly available on request. The ensemble will be uploaded to ESGF during 2020.

CMIP5 data is openly available via ESGF data nodes.

## Author contribution

Renate Wilcke and Erik Kjellström designed the structure and did most of the writing. Renate Wilcke did the analysis on temperature and three indices. Erik Kjellström performed the analysis on observations. Changgui Lin did the analysis on HWMI and WSDI. Anders Moberg did the analysis on the Stockholm time series. Daniela Matei provided the ensemble data from the MPI climate modelling group (as it was not openly available in the beginning of this study).

## Competing interests

The authors declare that they have no conflict of interest.

## Acknowledgements

The authors are grateful to Lennart Wern at SMHI for providing the monthly mean temperature data. The authors acknowledge the World Climate Research Programme's Working Group on Coupled Modelling, which is responsible for CMIP, and we thank the climate modelling groups (listed in Table S1) for producing and making available their model output. This study was supported by FORMAS (REGTREND).

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

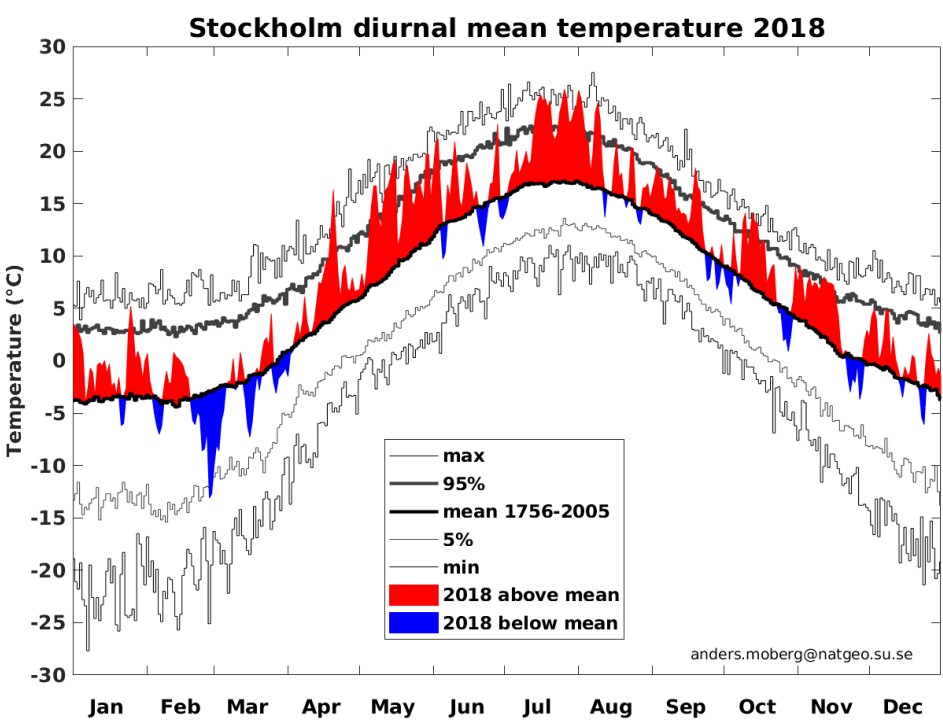

Figure 1: Diurnal mean temperatures in Stockholm 2018. For each day the anomaly with respect to the 250-year climatological mean for 1756-2005 is displayed in red (warm) or blue (cold). The diagram also shows the warmest and coldest diurnal mean temperature for each calendar day recorded within the 1756-2005 period as well as the corresponding 5th and 95th percentiles.

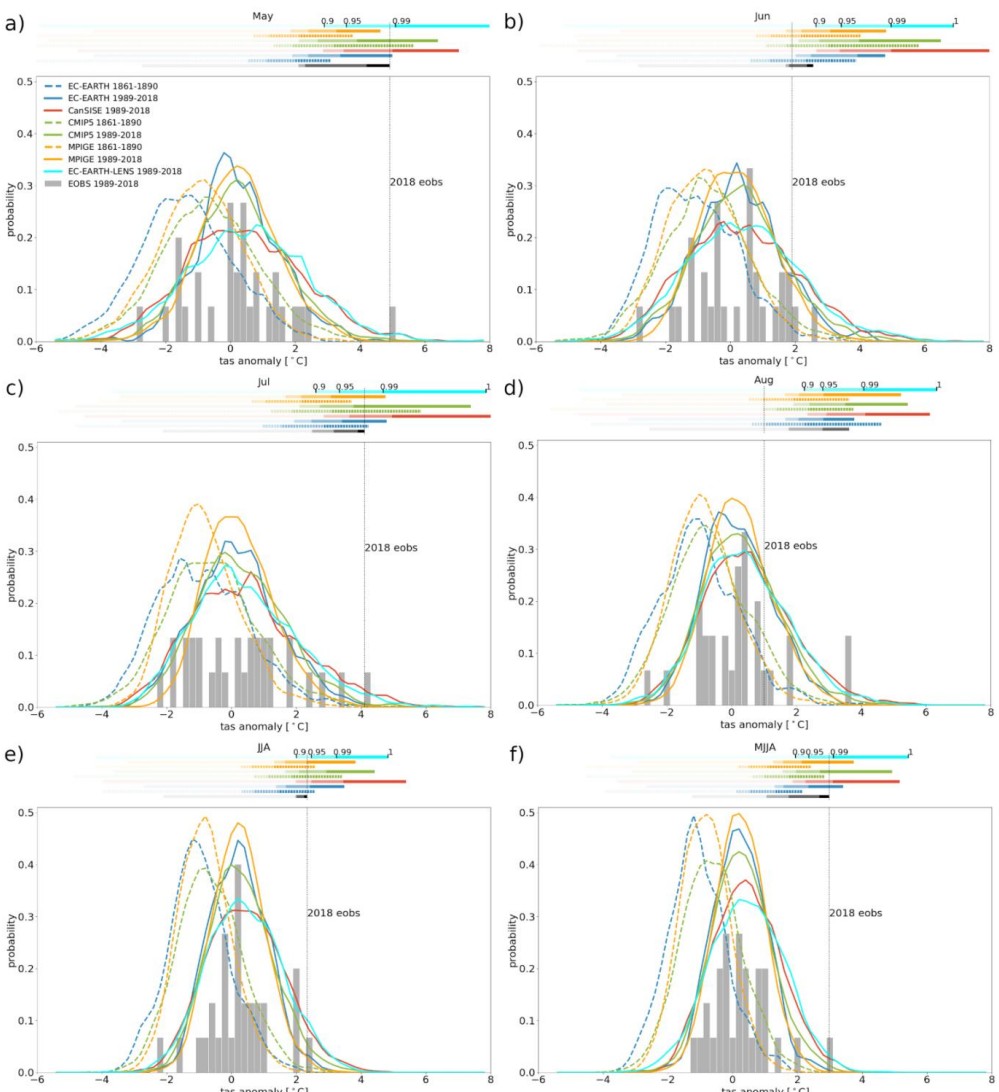

Figure 2. Probability distributions for monthly average temperature anomalies, calculated for 1861-1890 (dashed) and 1989-2018 (solid) against 1981-2010 for the southern half of Sweden, for the months of May, June, July, and August, and seasonally pooled for JJA and MJJA. The bars in the upper part of each panel are a guide to compare the positions of percentiles for each ensemble and period. The opacity of the bars indicates in steps the 90th, 95th, 99th, and 100 percentile ranges marked on the uppermost bar. Ensemble distributions are a kernel density fit, whereas the histogram for EOBS is based on actual data. The observed year 2018 for respective months/season is indicated by the vertical dotted black line.

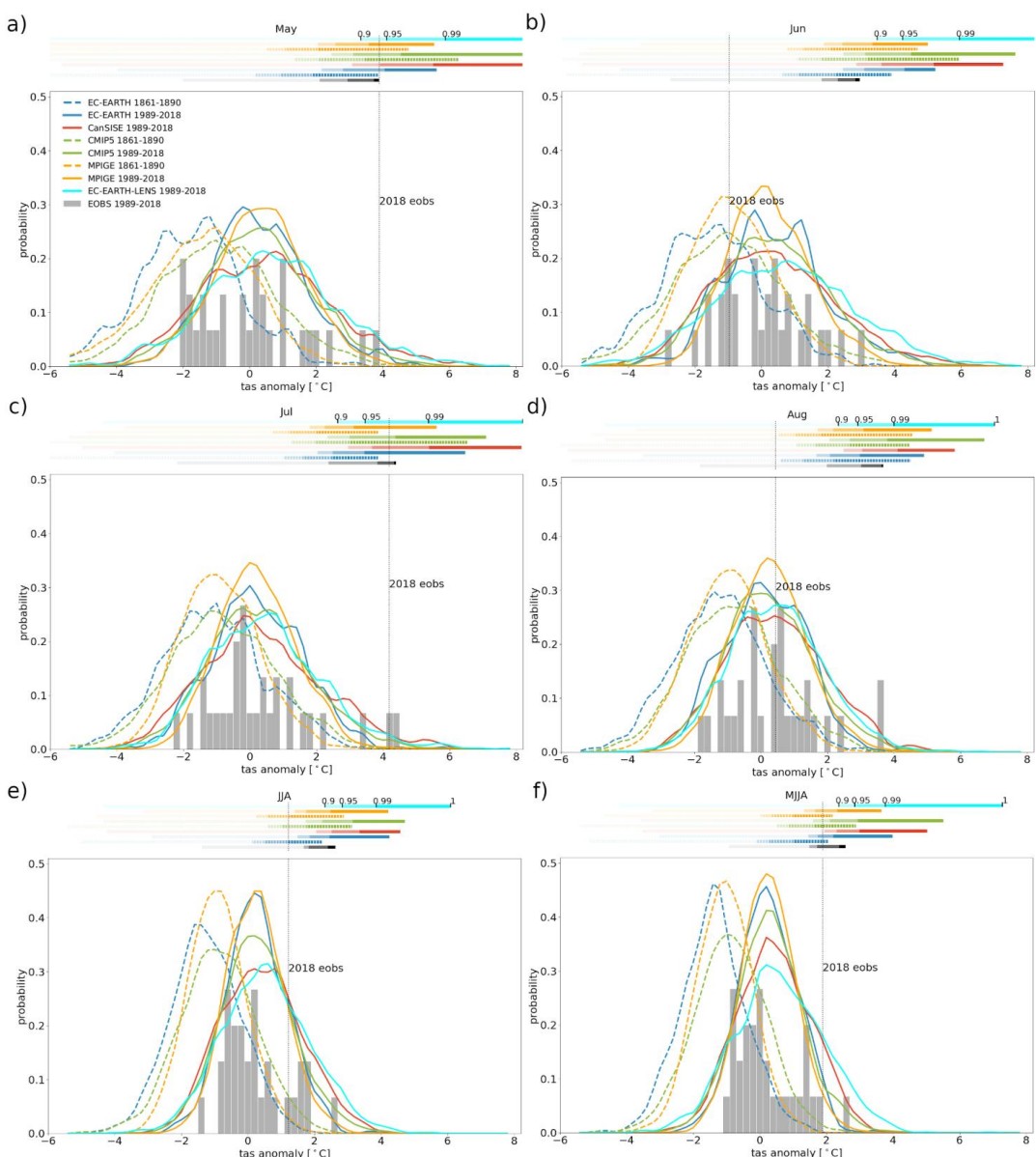

**Figure 3. As in Figure 2 but for the northern half of Sweden (as defined in Fig. S1 red).**

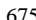

Figure 4. Left column: a) Total number of warm days (totWarmD), c) maximum number of consecutive warm days (max_conWarmD), and e) number of heat events (tot_events) for the period 1951-1980 (dashed) and 1989-2018 (dotted) for 4 models (x axis), calculated from the MJJA months and averaged over southern Sweden. All model members are pooled into one box. Corresponding data for E-OBS is shown to the right with the red mark indicating the median. Right row shows time series

**for each ensemble member (thin lines) and ensemble means (bold lines) for b) totWarmD, d) max_conWarmD, and f) tot_event. The number of days for 2018 as derived from E-OBS is indicated by the horizontal dashed line.**

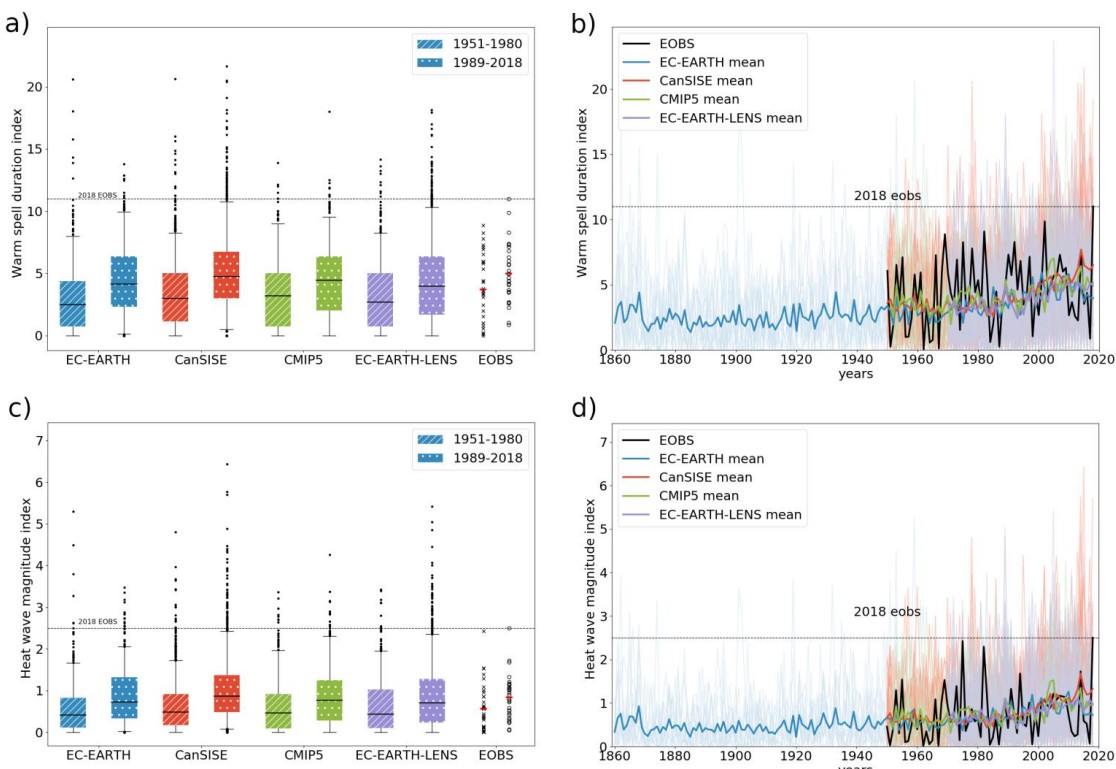

**Figure 5. As Figure 4 but for a, b) WSDI and c, d) HWMI.**

**Table 1. Observational data used in the study.**

| Data name | Period | Monthly or daily data | resolution |
|---|---|---|---|
| E-OBS v 19.0e | 1961—2018 | monthly/daily | 12.5 km |
| Sweden average | 1860—2018 | monthly | mean of 35 stations |
| Stockholm series | 1756—2018 | daily | point |

**Table 2. Climate model ensembles analysed in the study.**

| Ensemble name | CMIP | Period | Forcing rcp/ssp | Ensemble size | Monthly or daily data | GCM resolution |
|---|---|---|---|---|---|---|
| MPI-GE | 5 | 1861—2018 | 4.5 | 100 | monthly | ~1.8° |
| EC-EARTH (v2.3) | 5 | 1861—2018 | 8.5 | 16 | daily | T159L62 |
| EC-EARTH-LENS (v3.3.1) | 6 | 1970—2018 | SSP5-8.5 | 50 | daily | T255L91 |
| CanSISE | 5 | 1950—2018 | 8.5 | 50 | daily | T63 |
| CMIP5 r1 ensemble | 5 | 1861 / 1951—2018 | 8.5 | 78/15 | monthly/daily | |