# Peer review of "The extremely warm summer 2018 in Sweden - set in a historical context"

_Earth System Dynamics, 2020_

## Referee Comment (RC1) · Anonymous Referee #1 · 9 Jul 2020

**Review of "Wilcke, R. A. I., Kjellström, E., Lin, C., Matei, D., and Moberg, A.: The extremely warm summer 2018 in Sweden – set in a historical context"**

This manuscript uses long-term observations and 5 climate model ensembles to quantify how extreme, or how unusual the extended summer season of 2018 was in Sweden. The main result is that although all climate model ensembles include summers of similar warmth, they are quite to very rare (depending on which metric is considered), and in particular May 2018 as an individual month was exceptionally warm. Furthermore, the authors show the likelyhood of a summer as warm as 2018 is higher in recent decades compared to in pre-industrial times or in the mid 20$^{th}$ century. The manuscript is within scope of ESD and is based on sound methods. The conclusions which are drawn are supported by clear evidence. Overall, the English language is of a good standard. There are a few odd turns of phrase / constructions but I expect copy editing will be able to address these. Some aspects of the manuscript could be clearer and as such I suggest some minor revisions which I list below.

1. The single largest issue with this manuscript is that there are a large number of time periods which are quite confusing to follow at times. Related to this, it is unclear why the reference period (1981- 2010) and a recent past (1989–2018) are both required. Furthermore, it is hard to remember for which time periods each observational data set and each ensemble is available. I strongly encourage the authors to include a table with all of this information in one place. A table could also indicate which data sets / climate models are available for each of the 4 time periods presented at the start of section 2.
2. Line 33. There is no reference item for Räisänen (2018) in the reference list at the end.
3. Line 36. Suggest adding "temperature" after "average".
4. Line 36. Using the reference period 1961- 1990 adds to the confusion discussed in point 1 above. Could the 1981 – 2010 reference period be used instead ?
5. Lines 62 – 63. How does this study differ to those of Leach et al (2019) and Yiou et al (2019)? Please add some details to the introduction.
6. Lines 73 – 76.  Add details here to make it clear to a reader that only historical climate model simulations are considered – no long term future projections are considered here.
7. Lines 90 – 94. This information about the observations could be added to table 1. Please add details if daily or monthly values from E-OBS were used here.
8. Line 98. typo. "complement of correct" → complement or correct.
9. Line 134. Again this relates to point 1 above. Here it is stated "we analyse 294 simulations adding up to 8820 summers over a 30 year period." Which 30-year period? Or multiple 30-year periods? Please clarify the text here.
10. Lines 138-139. From which observational data sets and climate models are daily maximum and daily averages taken? This information is partially given in Table 1 but it is not clear which data was used from the CMIP5 multi-model ensemble. Furthermore, are the results comparable if for some datasets / ensembles daily values are used whereas for others monthly values are used?
11. Line 144 – 146. The definition of a warm day here is not completely clear. Specifically, is the 98$^{th}$ percentile threshold for each grid point the same for all days i.e. no time dependence between May – August? Please clarify this.
12. Section 2.3. It is quite hard to understand all of these diagnostics. Please explain what the differenes are between the WSDI and the warm days as defined by equation 1. Please also revise the explanation of the HWMI in lines 162 – 167 as currently I cannot understand this.
13. Line 173. Add the years after "reference period" to remind a reader.
14. Line 178. Add "temperature" before "data".
15. Line 194. Add that the 95$^{th}$ percentile referred to here is calculated from the 1756 – 2005 Stockholm temperature data. This is another example of yet another different averaging period (see point 1 above).

16. Line 194. It would be interesting to know which other year exceeded the 95$^{th}$ percentile. Could this information be added to the text?
17. Line 214. This paragraph needs to include what data the numbers / results discussed here are based on which I think is E-OBS.
18. Line 231, Add the relevant years after "For the recent past period (…).."
19. Line 249. Given that there are only 5 figures in this manuscript, Figure S3 could be moved from the supplement to the main article.
20. Line 268. Please clarify what is meant by "… the chosen fit".
21. Figure 2. Please expand the explanation in the caption of the colorbars at the top of each panel.

---

## Referee Comment (RC2) · Anonymous Referee #2 · 16 Jul 2020

This study provides a statistical analysis of summer temperature-based heat wave metrics in several different types of observations and several different ensembles of climate models (both multi-model and initial condition ensembles) in order to contextualize the 2018 heat event in Sweden. I commend the authors for their thorough characterization of the event; many aspects are evaluated including conditions in northern vs. southern Sweden, daily average temperature, monthly and seasonally aggregated temperature, warm day indices, a heatwave magnitude index, and a heatwave duration index. Authors find that the 2018 heat event was noteworthy within the historical station record due to the number of above-climatological-average days in the May-August season, rather than the magnitude of the temperature anomalies on those above-average days. However, the station data used has a summer correction applied to it before conclu-

sions are made, and it is not clear how this methodological choice influences results.

Additionally, the large number of model simulations used in the second part of the study are not described in sufficient detail. For instance, which models make up the CMIP5 ensemble used in this study? What component versions are used in the CMIP5 vs. CMIP6 versions of EC-Earth? Certain models may have systematic biases that influence to what extent the 2018 Swedish summer is considered exceptional; biases that would be useful to consider and discuss. Because of methodological choices (different spatial resolutions and land masks, corrections to observational datasets, assorted radiative forcing pathways), it is somewhat challenging to interpret the comparisons between models and observations. I believe addressing methodological inconsistencies and evaluating sensitivities to other methodological choices (i.e., base period) would strengthen this paper considerably.

General thoughts:

Clarity is also undermined by numerous inconsistencies in capitalization, spelling, tense, and sentence structure. I have indicated several common issues below, but significant language editing should be carried out before resubmission.

Persistent blocking conditions are identified throughout the study as the driver of the 2018 Swedish heatwave but are not systematically evaluated. Assessing whether the models you evaluate are able to simulate these synoptic conditions would strengthen the study by ensuring that the longer duration, higher intensity events seen in the models originate from similar-to-observed weather patterns. This would begin to address the question of whether global climate models are able to realistically simulate Swedish heat events.

Specific comments:

L10 and throughout: There are several inconsistencies with the presence and absence of hyphens (i.e., long-lasting vs. long lasting, MPI-GE vs. MPIGE).

L11: A comma is missing in the list.

L13: the whole of Sweden?

L15: What heatwave indicators are used?

L22 and throughout: Sentences of this structure (dependent clause before independent clause) need a comma separating the two clauses.

L23: Identifying the models by '1861-90' and '1951-80' time ranges is unclear at this point. Are 1861 and 1951 start years of different models? What are the ranges indicating in this case?

L40: and rather than ;

L47: References to support this statement would be useful.

L51-52: Variability on what time scales? What is the time scale of a warm period?

L59: Large year-to-year variability of what?

L66: 'Leach et al.' is missing the year.

L75-76: What does "to what extent . . . may have changed" refer to? Duration? Intensity? Frequency?

L98: What does "used to complement of correct" mean?

L110: Wouldn't these adjustments affect your analysis in fundamental ways?

L126: from 2006 onward,

L131: Can you comment on differences between the RCP and SSP forcing scenarios used? As I understand it, RCP8.5 and SSP585 are similar but not interchangeable.

L140: Does "pooled" mean seasonally averaged?

L142: "were" is inconsistent with the "is" in the prior paragraph. Either present or past

tense should be used consistently throughout the methods section.

L154: Is this threshold the threshold stated above?

L182: How is the diurnal mean temperature computed?

L188 and throughout: You switch between northern and Northern, southern and Southern, etc. I think the lower case monikers (as you have used here) are correct. For cardinal direction, capitals should be used.

L192-193: Are results sensitive to your choice of the climatological period?

L198-199, 201: Because the figures you are mentioning here are not a part of this paper, it is confusing to list them as Fig. 5 and 10 etc.

L201: What does a heatwave intensity of 65 K mean?

L206: The two opening clauses are repetitive.

L288-289: The legend of Figure 3B seems to cover some of the data.

L391: Can these 1 in 20 and 1 in 100 statistics also be discussed in the results section?

L400: To occur in models?

Figure S1: Models and observations appear to have different spatial coverage; are results affected by this? It tends to be standard in model intercomparisons to re-grid models to a common grid.

Figure 2: The color differences corresponding to percentiles in the bars above the panels are difficult to see. Can you describe the kernel density fit used in more detail as well?

---

## Author Comment (AC1) · 2 Sep 2020

**Response to RC1**

We thank reviewer 1 for the constructive comments. Our response is written in blue and located behind each reviewer comment.

**Review of "Wilcke, R. A. I., Kjellström, E., Lin, C., Matei, D., and Moberg, A.: The extremely warm summer 2018 in Sweden – set in a historical context"**

This manuscript uses long-term observations and 5 climate model ensembles to quantify how extreme, or how unusual the extended summer season of 2018 was in Sweden. The main result is that although all climate model ensembles include summers of similar warmth, they are quite to very rare (depending on which metric is considered), and in particular May 2018 as an individual month was exceptionally warm. Furthermore, the authors show the likelyhood of a summer as warm as 2018 is higher in recent decades compared to in pre-industrial times or in the mid 20th century. The manuscript is within scope of ESD and is based on sound methods. The conclusions which are drawn are supported by clear evidence. Overall, the English language is of a good standard. There are a few odd turns of phrase / constructions but I expect copy editing will be able to address these. Some aspects of the manuscript could be clearer and as such I suggest some minor revisions which I list below.

1. The single largest issue with this manuscript is that there are a large number of time periods which are quite confusing to follow at times. Related to this, it is unclear why the reference period (1981- 2010) and a recent past (1989–2018) are both required. Furthermore, it is hard to remember for which time periods each observational data set and each ensemble is available. I strongly encourage the authors to include a table with all of this information in one place. A table could also indicate which data sets / climate models are available for each of the 4 time periods presented at the start of section 2. - We have removed one reference period (1961-1990) and further added a table about observational data sets to the manuscript. The data availability is now clear from Table 1 (observational data) and Table 2 (model data). The 30-year periods used for analysis are stated clearly in L80-81. We use one reference period, which is the new reference period defined by WMO, 1981-2010. The recent past or present day climate period is relevant as we are studying extreme temperature events in the recent past. Excluding the years beyond 2010 would give an out-dated picture of the available climate knowledge about extreme temperatures. We therefore argue, to keep both periods.

2. Line 33. There is no reference item for Räisänen (2018) in the reference list at the end. - Now included: Räisänen J. (2018). Energetics of interannual temperature variability. Clim. Dyn. 52: 3139–3156. DOI: 10.1007/s00382-018-4306-0

3. Line 36. Suggest adding "temperature" after "average". - We agree and add "temperature" to the sentence.

4. Line 36. Using the reference period 1961- 1990 adds to the confusion discussed in point 1 above. Could the 1981 – 2010 reference period be used instead ? - We

understand the possibility of confusion here and therefore will use only one reference period, 1981-2010.

5. Lines 62 – 63. How does this study differ to those of Leach et al (2019) and Yiou et al (2019)? Please add some details to the introduction. - Both studies are pure attribution studies. So we refer to their results as they underpin our conclusions drawn from simpler statistics without a factual-counterfactual approach. The main differences are the metrics which have been analysed, meaning different aspects of the heat event. Additional differences are how the heat event is defined, for which region the analysis was undertaken and what ensembles have been used.
   - Yiou et al (2019) analyse a 19-day window in July 2018 over a larger Scandinavian domain. Using regional climate model ensembles as well as a CMIP5 ensemble they do an unconditional and a conditional attribution approach to demonstrate the thermodynamic contribution of human-induced climate change to describe the probability and intensity of the summer 2018 event in Scandinavia.
   - Leach et al. (2019) analysed the return values of annual maximum of the 1-, 10-, and 90- day running mean of daily mean 2-m temperature for the factual and counterfactual simulations of HadGEM3-A global atmospheric model.

   We add to the introduction text in L63: "Determining the probability of an event like 2018 being observed, plus the probability of occurrence conditioned on large scale patterns, Yiou et al. (2019) concluded that the probability of an event such as the one observed in the second half of July 2018 has increased as a result of human-induced climate change. "
   Further in L69 we change the sentence to "The aim of the study is to expand findings from previous studies by evaluating a broad range of temperature conditions in Sweden during summer 2018 in relation to the historical climate."

6. Lines 73 – 76. Add details here to make it clear to a reader that only historical climate model simulations are considered – no long term future projections are considered here. - Changed to: "In a next step we investigate the likelihood of such a summer to have occurred in the past century using five large global climate model ensembles, some of which are covering a period since 1860, and others starting in the second half of the 20th century, up to 2018."

7. Lines 90 – 94. This information about the observations could be added to table 1. Please add details if daily or monthly values from E-OBS were used here. - We use daily and monthly data from E-OBS, depending on the kind of analysis step we were at. This has been clarified now in the sentence in L90: " i) the gridded daily and monthly climatology E-OBS version 19.0e …" We also added a table to give an overview over the observational data used.

8. Line 98. typo. "complement of correct" → complement or correct. - Corrected

9. Line 134. Again this relates to point 1 above. Here it is stated "we analyse 294 simulations adding up to 8820 summers over a 30 year period." Which 30-year period? Or multiple 30- year periods? Please clarify the text here. - The 8820 summers are for each 30 year periods (294 * 30 = 8820).
This has been written more clearly now, "In total, we analyse 294 simulations expanding the sample size for each 30 year period to 8820 (30 summers times 294 simulations)."

10. Lines 138-139. From which observational data sets and climate models are daily maximum and daily averages taken? This information is partially given in Table 1 but it is not clear which data was used from the CMIP5 multi-model ensemble. - We added a table (Table S1) to the supplement which lists all CMIP5 simulations used here. In the comparison with climate models we used E-OBS only as observational data.
Furthermore, are the results comparable if for some datasets / ensembles daily values are used whereas for others monthly values are used? - The monthly and daily analysis are meant to complement each other, rather than being strictly compared. The monthly mean is a good indicator for a larger heat event as a single heat day doesn't affect the monthly average as much. Another reason was that we wanted to use the MPIGE which, with its 100 members, is the largest ensemble, so far. The caveat is that only monthly data had been made available. For each metric we calculated and analysed we used the same temporal resolution for each ensemble. That means, analysis based on daily data does not include MPIGE.

In L138 we write: "All analysis is based on daily and monthly temperature data." It continues further with "To assess the average temperature for the summer season we use monthly mean temperatures for four individual summer months (May, June, July, and August), separately and for two summer seasons (JJA, MJJA; averaged)." Here we use only the monthly data as stated.
"Furthermore, three warm day indices, based on daily values, awere used to assess the temperature variability during summer:"
At the end of the section we add: "The analysis is carried out for all model ensembles where respective data is available. I.e. the pdf analysis based on monthly data includes all five ensembles (c.f. Table 2 and Table S1), whereas the warm day indices are based on daily temperature values which are not available from MPI-GE."

11. Line 144 – 146. The definition of a warm day here is not completely clear. Specifically, is the 98th percentile threshold for each grid point the same for all days i.e. no time dependence between May – August? Please clarify this. - For each grid point we calculate the 95th percentile from all days within the 4 month period of May to August. Those 95th percentiles are then used as thresholds to define the warm days at each grid point. That means, the relative threshold is the same for all grid points (95th percentile). Sweden is a rather long country covering a big range of latitudes. A fixed threshold value for all grid points would lead to no heat days in the Northern part, though locally experienced and affecting the environment there had been heatwaves, relative to the average temperatures there.

We changed the sentence to: "For simplicity reasons we define a warm day as a day i with T_(max,i) greater than a relative threshold. The threshold is the 95th percentile calculated of all  T_max in May to August (MJJA) during the reference period (see Eq. (1))."

12. Section 2.3. It is quite hard to understand all of these diagnostics. Please explain what the differenes are between the WSDI and the warm days as defined by equation 1. Please also revise the explanation of the HWMI in lines 162 – 167 as currently I cannot understand this. - We restructured the section partly which reads now:

"All analysis is based on daily and monthly temperature data. We look at both daily average and daily maximum temperatures, as well as monthly means of daily average temperature. To assess the average temperature for the summer season we use monthly mean temperatures for four individual summer months (May, June, July, and August) separately and for two summer seasons (JJA, MJJA; seasonal averaged). Temperature anomalies are calculated against the reference period 1981–2010.

Furthermore, three warm day indices, based on daily values, are used to assess the temperature variability during summer:. The indices based on equation 1 are i) the "total number of warm days per year" (totWarmD), ii) the "maximum number of consecutive warm days per year" (max_conWarmD), iii) the "number of heat events" (tot_event), where an event is defined as minimum 3 consecutive days of T_max>threshold.

For simplicity reasons we define a warm day as a day i with T_(max,i) greater than a relative threshold. T, where the threshold is the 95th percentile calculated of all of MJJA T_max in May to August (MJJA) during the reference period (see Eq. (1)).

$$T_{(max,i)} > p95(T_{(max,(MJJA\ 1981-2010))})$$
$$(1)$$

This simple definition based on a relative measure is chosen to make it possible to compare conditions in different parts of Sweden to each other. For example, a perceived heat wave in the colder North of Sweden, may not even appear in an analysis involving an absolute threshold representative of conditions in southern parts of the country like, e.g., summer days defined as days with  T_max  >25° C. Other examples of benefits with a relative measure involve comparison of coastal and inland stations or between low and high altitude stations for similar reasons.

Additionally we calculated two heat wave indices commonly used. The Warm Spell Duration Index (WSDI, Orlowsky and Seneviratne, 2011) that can be compared to max_conWarmD, that differ in their definition of the warm day. The WSDI is calculated based on an individual threshold for each day of the year (doy). A warm day is defined as a day with T_(max,i) larger than the 90th percentile of A_i, as defined in Eq. 2 (from Eq. 1 in Russo et al. 2014) .

$$A_i = \bigcup_{y=1981}^{2010} \bigcup_{d=i-15}^{i+15} T_{max,y,d} \qquad\qquad (2)$$

Where ∪ denotes the union of sets and $T_{(max,y,d)}$ is the daily maximum temperature of the day d in the year y. The WSDI calculates then the maximum number of consecutive warm days (larger than 3), i.e. for a given year (or season), the WSDI is the longest duration of any such heat-wave event.
The second heat-wave index is the Heat Wave Magnitude Index (HWMI, Russo et al. 2014), which uses the same warm day definition as WSDI. Whereas the WSDI takes into account only the duration, the HWMI also considers the magnitude of the heat-wave. The HWMI takes into account multiple sub-maximum temperatures of an event by summing them up and mapping them to a probability (called magnitude) related to annual maximum magnitudes of the reference period. By that it weights the duration more than the absolute maximum temperature of an event. This relates to the heat stress which builds up with many warm days in a row rather than a couple of very warm days in a row. A more detailed description of how to calculate the HWMI can be found in Russo et al. (2014)."

13. Line 173. Add the years after "reference period" to remind a reader. - Added.

14. Line 178. Add "temperature" before "data". - Added.

15. Line 194. Add that the 95th percentile referred to here is calculated from the 1756 – 2005 Stockholm temperature data. This is another example of yet another different averaging period (see point 1 above). - We understand that we use many different periods, which we tried to give a better overview in Table 1 and 2 as well as in the text: "The historical context is given by comparing observed conditions in 2018 to observed and simulated climate conditions for: i) a pre-industrial period 1861–1890, ii) a mid 20th century period 1951–1980 and iii) our reference period 1981–2010. For some analysis we also look at the most recent past 30-year period ending 2018 (1989–2018) partly overlapping the reference period. For the longest possible time perspective, we also consider the period 1756–2018 using the Stockholm temperature series." However, the advantage and reason why we include the Stockholm temperature series, is its exceptional length. With comparing 2018 to only 1981-2010 we would miss out on the opportunity to compare an extreme summer to the past 250 years.
We edited the sentence: "For the full four-month period MJJA we note that more than 35 % of the days were above the long-term (1756–2005) climatological 95th percentile calculated for each day (upper grey line in the figure)."

16. Line 194. It would be interesting to know which other year exceeded the 95th percentile. Could this information be added to the text? - The only other year that has more than 20% days above the 95th percentile is 2002. We add that information to the text: "Only one additional year, the year 2002, exceeds the 95th percentile with 20 % of the days in a full May−August season."

17. Line 214. This paragraph needs to include what data the numbers / results discussed here are based on which I think is E-OBS. - Correct, the observations used here were the E-OBS data set. Edited to "Probability distributions (pdfs) of the four single

summer months (May, June, July, and August) from E-OBS temperature data in Fig. 2 (a–d) further illustrate the rarity of the weather situation in 2018 in Southern Sweden. "

18. Line 231, Add the relevant years after "For the recent past period (…).." - Added.

19. Line 249. Given that there are only 5 figures in this manuscript, Figure S3 could be moved from the supplement to the main article. - We agree with this suggestion and move Figure S3 to the manuscript, changing it to Figure 3 and changing former Figures 3-4 accordingly.

20. Line 268. Please clarify what is meant by "… the chosen fit". - The "chosen fit" refers to the way the distribution is calculated and presented. For E-OBS as "one-member ensemble", we chose to show a histogram which best represents how the temperature anomalies are distributed. For each ensemble we pooled together all ensemble members and calculated the kernel density estimate with a gaussian fit to gain a smooth curve. It has a strong visual aspect here, as nine histograms wouldn't be distinguishable in one figure. However, the density function curves go asymptotically to zero probability in their tails. When calculating the 100th percentile (indicated by 1 on top of the bars in Fig. 2) this value can be larger than the empirical value, meaning that it can be outside of the actual value range provided by the ensemble members.
We edit Line 268 and add: "This also indicates that CMIP5 temperatures like in May 2018 fall only into the distribution because of the chosen fit of the Gaussian pdf estimate. (The pdf curve continues smoothly to zero even if there is no value at the end of the tail.)"

21. Figure 2. Please expand the explanation in the caption of the colorbars at the top of each panel. - The caption reads now "Probability distributions for monthly average temperature anomalies, calculated for 1861-1890 (dashed) and 1989-2018 (solid) against 1981-2010 for the Southern half of Sweden, for the months of May, June, July, and August, and seasonally pooled for JJA and MJJA. The bars in the upper part of each panel are a guide to compare the positions of percentiles for each ensemble and period. The opacity of the bars indicates in steps the .9, .95, .99, and 1 percentile ranges marked on the uppermost bar. Ensemble distributions are a kernel density fit, whereas the histogram for EOBS is based on actual data. The observed year 2018 for respective months/season is indicated by the vertical dotted black line."

---

## Author Comment (AC2) · 2 Sep 2020

This study provides a statistical analysis of summer temperature-based heat wave metrics in several different types of observations and several different ensembles of climate models (both multi-model and initial condition ensembles) in order to contextualize the 2018 heat event in Sweden. I commend the authors for their thorough characterization of the event; many aspects are evaluated including conditions in northern vs. southern Sweden, daily average temperature, monthly and seasonally aggregated temperature, warm day indices, a heatwave magnitude index, and a heatwave duration index. Authors find that the 2018 heat event was noteworthy within the historical station record due to the number of above-climatological-average days in the May-August season, rather than the magnitude of the temperature anomalies on those above-average days. However, the station data used has a summer correction applied to it before conclusions are made, and it is not clear how this methodological choice influences results. - The summer correction does change the results only marginally and does not affect the conclusions to draw from them. In the manuscript we added in L118 "This adjustment does alter our results only marginally and does not affect the conclusions we draw."

Additionally, the large number of model simulations used in the second part of the study are not described in sufficient detail. For instance, which models make up the CMIP5 ensemble used in this study? - We have added a table to the supplement listing the CMIP5 models used in this study, Table S1

What component versions are used in the CMIP5 vs. CMIP6 versions of EC-Earth? - We use EC-EARTH v2.3 (https://view.es-doc.org/?renderMethod=id&project=cmip5&id=72138708-d53e-11df-938c-00163e9152a5&version=5&client=esdoc-search) and EC-EARTH v3.3.1 (https://explore.es-doc.org/cmip6/models/ec-earth-consortium/ec-earth3). The information about EC-EARTH versions is now added to Table 2 (Table 1 is now a table about observational data used in this study).

Certain models may have systematic biases that influence to what extent the 2018 Swedish summer is considered exceptional; biases that would be useful to consider and discuss. Because of methodological choices (different spatial resolutions and land masks, corrections to observational datasets, assorted radiative forcing pathways), it is somewhat challenging to interpret the comparisons between models and observations. - Yes, all those GCMs have their systematic biases. We account for those by using percentile thresholds for index calculations. If a model is systematically 1 K too warm in summer and assuming this bias is stationary, it would not affect our analysis as we were looking at, e.g., warm days with temperatures above the 95th percentile. This threshold is calculated for each ensemble respectively. The same holds true for the analysis of temperature anomalies, as we did not

compare the absolute values to the observations but calculated the anomalies for each ensemble respectively. So, what we compare to observations is the change in anomalies and the change for certain indices over time. Where the change again is calculated within each ensemble.
When we looked into the mean temperature bias (difference to E-OBS), we found EC-EARTH (KNMI ensemble), MPIGE, and CanESM with a cold bias for MJJA months, ranging from -1 K to -0.2 K.
A full analysis on the various moments of biases of all GCM ensembles are out of scope for this study and would rather be worth its own publication. We are aware of that limitation and address it at the end of section 3 the following way:
"All models suffer from simulation biases. For summer temperatures all models show a certain cold bias over northern Europe which range from -0.2 K to -1 K (not shown). However, as the analysis is done for each ensemble separately and focussed on anomalies, the impact of any systematic bias is reduced. "

I believe addressing methodological inconsistencies and evaluating sensitivities to other methodological choices (i.e., base period) would strengthen this paper considerably. General thoughts: Clarity is also undermined by numerous inconsistencies in capitalization, spelling, tense, and sentence structure.  - We have corrected all inconsistencies we found.
I have indicated several common issues below, but significant language editing should be carried out before resubmission. Persistent blocking conditions are identified throughout the study as the driver of the 2018 Swedish heatwave but are not systematically evaluated. Assessing whether the models you evaluate are able to simulate these synoptic conditions would strengthen the study by ensuring that the longer duration, higher intensity events seen in the models originate from similar-to-observed weather patterns. - Thank you for the remark. We add a brief discussion in the manuscript:
"All models suffer from simulation biases. For summer temperatures all models show a certain cold bias over northern Europe which range from -0.2 K to -1 K (not shown). However, as the analysis is done for each ensemble separately and focussed on anomalies, the impact of any systematic bias is reduced.
It is known that GCMs have difficulties to capture many aspects of blocking (e.g., Davini and D'Andrea, 2016 ; Dawson and Palmer, 2016). Studies on atmospheric blocking representation in the GCMs used here, agree on an underestimation of blocking frequency mostly in winter, but also in summer (EC-EARTH: Hartung et al. (2017); CMIP5: Woolings et al. (2018), Masato et al. (2013), Dunn-Sigouin and Son (2013); MPI-GE: Maher et al. (2019), Müller et al. (2017); CanESM: Schaller et al. (2018), Brunner et al. (2017)).
Increasing horizontal and vertical resolution of GCMs can strongly improve the ability to capture atmospheric blockings (Hartung et al. (2017), Jung et al. (2012)). However a positive effect of increased resolution could not be confirmed for summer blockings (Schiemann et al., 2017). Sousa et al. (2018) found that the frequency of summer blocking (23% of the days, and only 9% over Scandinavia) is less than of winter blocking (35% of the days). The small sample size of summer blockings over northern Europe handicap statistical analysis, hence the amount of references on that topic is low.
For this study we focus on the advantage of large ensembles that increase the sample size for statistical evaluation of high temperature events. "

This would begin to address the question of whether global climate models are able to realistically simulate Swedish heat events. Specific comments:

L10 and throughout: There are several inconsistencies with the presence and absence of hyphens (i.e., long-lasting vs. long lasting, MPI-GE vs. MPIGE). - Corrected.

L11: A comma is missing in the list. - Added.

L13: the whole of Sweden? - Corrected

L15: What heatwave indicators are used? - The heatwave indicators are listed in section 2.3. We find such a list does not fit in the abstract and suggest to just add "five" before heat wave indicators, to indicate that we look at multiple and that they are too many to list.

L22 and throughout: Sentences of this structure (dependent clause before independent clause) need a comma separating the two clauses. - We changed that sentence to: "For all indices evaluated, we find that probabilities of a summer like in 2018 have increased from relatively low values in the pre-industrial (1861–1890, one ensemble) as well as the recent past (1951–1980, all five ensembles) to the most recent decades (1989–2018). " And edited similar sentences according to your suggestion.

L23: Identifying the models by '1861-90' and '1951-80' time ranges is unclear at this point. Are 1861 and 1951 start years of different models? What are the ranges indicating in this case? - See above. We hope changing the sentence in that way makes our statement more clear. One ensemble with pre-industrial data showing an increase in probability for summers like 2018. All five ensembles (of which four start 1951) show an increase in this probability from 1951 to 2018.

L40: and rather than ; - we changed the punctuation marks from twice ";" to "," and ", and". The sentence reads now: "The hot and dry conditions in summer 2018 in Sweden were associated with severe consequences for people and the environment including: health problems and excessive mortality rate among people (Åström et al. 2019), water shortages with adverse implications for arable land and pastures (Buras et al. 2019) including lack of forage, and unusual large areas affected by forest fires (Krikken et al. 2019). "

L47: References to support this statement would be useful. - It is not clear to us which statement is meant. L47 reads: "As a consequence of global warming, heat waves have become more frequent and intense (e.g. SREX 2012; IPCC 2018; Sippel et al 2020)." Which already includes the reference to IPCC 2018 and Sippel et al. 2020.

L51-52: Variability on what time scales? What is the time scale of a warm period? - We decided to remove that sentence as in this study we won't discuss the different natures of heat-waves.

L59: Large year-to-year variability of what? - Meant is the large year-to-year variability of heat wave occurrence over Scandinavia. However, we removed that sentence.

L66: 'Leach et al.' is missing the year. - Year added.

L75-76: What does "to what extent . . . may have changed" refer to? Duration? Intensity? Frequency? - The two sentences before read now like: "Different aspects of heat-wave characteristics; including number of heat events, total number of warm days, total number of consecutive days and heat-wave intensity, are assessed. In a next step we investigate the likelihood of such a summer to have occurred in history using five large global climate model ensembles, some of which are covering a period since 1860, and others starting in the second half of the 20th century, up to 2018. " And give an indication that we are investigating the duration (total number of warm days, total number of consecutive warm days), the intensity (heat wave intensity (HWMI)) as well as the frequency (number of heat events) of heat-waves occurring. That together we sum up as "extent" of a heat-wave. The change refers to the analysis of different periods in the past and present.

L98: What does "used to complement of correct" mean? - This is a typo. Correct it says: "For stations with missing data, mostly in the first decades, and for stations where inhomogeneities have been identified (following Alexandersson and Moberg (1997) and Moberg and Alexandersson (1997)), data from surrounding stations have been used to complement or correct the temperature series."

L110: Wouldn't these adjustments affect your analysis in fundamental ways? - In addition to using data that are adjusted both before 1859 (for a supposed bias due to poorly protected thermometers, in the early instrumental period, EIP) and after 1870 (for the urban warming trend), we repeated the analysis of the Stockholm record without these adjustments. We find that the results are only marginally affected by the choice of variant. The characteristics of May-August season in 2018 is as outstanding regardless which variant of Stockholm data was used. We mention that now in the manuscript too in L118: "This adjustment does alter our results only marginally and does not affect the conclusions we draw."

L126: from 2006 onward, - changed accordingly.

L131: Can you comment on differences between the RCP and SSP forcing scenarios used? As I understand it, RCP8.5 and SSP585 are similar but not interchangeable. - Yes, correct. RCPs and SSPs are not the same, obviously. RCPs are four climate pathways defined by radiative forcing at the end of the century. Whereas SSPs are five socioeconomic development trajectories defined in terms of challenges to adaptation and mitigation and are not matched to reference RCPs. SSP5 is the conservative trajectory with continued fossil fuel usage. Meinshausen et al. (in review 2019) write "The SSP5-8.5 marks the upper edge of the SSP scenario spectrum with a high reference scenario in a high fossil-fuel development world throughout the 21st century". The "8.5" in the name "SSP5-8.5" refers to the approximated radiative forcing in 2100 and would be about the same as in RCP 8.5.
It is correct that SSP5-8.5 and RCP 8.5 do not provide the same forcing, but SSP5-8.5 is the one closest to RCP 8.5. In particular in the beginning of the scenario period the differences are neglectable as they are much lower than the model variability. We use only the years up to 2018.

In the manuscript we add the following sentence to clarify: "Though SSP5-8.5 is not identical to RCP 8.5 forcing, it is the trajectory which is closest to RCP 8.5 pathway (Meinshausen et al., in review 2019)."

L140: Does "pooled" mean seasonally averaged? - With pooled we mean that we group, e.g., the four months May, June, July, August together, that we consider all four months together in one analysis. It is not a seasonal average. We edited the sentence to make this more clear: "To assess the average temperature for the summer season we use monthly mean temperatures for four individual summer months (May, June, July, and August) separately and for two summer seasons (JJA, MJJA; not averaged)."

L142: "were" is inconsistent with the "is" in the prior paragraph. Either present or past tense should be used consistently throughout the methods section. - We corrected the tense in the method section and the whole manuscript.

L154: Is this threshold the threshold stated above? - Yes, it is defined in L145 and equation 1.

L182: How is the diurnal mean temperature computed? - The diurnal mean temperature in Stockholm is calculated as described in Moberg et al. (2002), which has been referenced to in L107, when the dataset was introduced in the text.
The number of observations per day and the timepoints of the day when temperatures were observed have changed several times since the start in 1756. Moberg et al. (2002) presented an attempt to estimate diurnal mean temperatures that always should account for the observation scheme.
After 1859, the calculation method is the same as the ones that SMHI has used for all stations in its station network (explained in Swedish at the SMHI web site: https://www.smhi.se/kunskapsbanken/meteorologi/hur-beraknas-medeltemperatur-1.3923). Since 1947 the following formula is used, $Tm=(aT07+bT13+cT19+dTx+eTn)/100$, where T07, T13, T19 are the temperature observations at UTC 06, 12, 18 and Tx and Tn are the diurnal max and min temperature. Thus, the diurnal mean temperature is a weighted average of temperatures calculated at three fixed timepoints and the daily Tx and Tn. The weights a,b,c,d,e are determined for different longitudes and thus differ among sites in the SMHI network. Moberg et al. (2002) use the same weights for their Stockholm record as SMHI uses for data from the same station.

L188 and throughout: You switch between northern and Northern, southern and Southern, etc. I think the lower case monikers (as you have used here) are correct. For cardinal direction, capitals should be used. - Thank you for pointing that out. We edited the manuscript and followed the explanation given here https://dictionary.cambridge.org/de/grammatik/britisch-grammatik/east-or-eastern-north-or-northern.

L192-193: Are results sensitive to your choice of the climatological period? - The number of days with temperatures above a certain percentile is certainly affected to some extent by the choice of climatological period. However, for Figure 1 where we use the long Stockholm record it is possible to use a very long period to define the percentile. This has both the advantage of

including a long time period well before the industrial era and also increasing the sample size which causes the curve of daily percentile to be much smoother and with fewer irregularities. It is more relevant to look at how sensitive the results are to the choice of variant of the Stockholm temperature series. Thus, in addition to using data that are adjusted both before 1859 (for a supposed bias due to poorly protected thermometers, in the early instrumental period, EIP) and after 1870 (for the urban warming trend), we repeated the analysis of the Stockholm record without these adjustments. We find that the results are only marginally affected by the choice of variant. But in all cases, 2018 stands out as remarkable as it has about 35%, or slightly more, days above the 95th percentile in the MJJA season , while the year on the second place has just a little above 20% of the days being warmer than the long-term 95th percentile.

L198-199, 201: Because the figures you are mentioning here are not a part of this paper, it is confusing to list them as Fig. 5 and 10 etc. - We are referring to figures in a reference paper, and want to guide the reader to the relevant figures we compare our results with. We suggest therefore to continue to mention the figures in the sentence, but we try to be more clear by writing it this way: "Hoy et al. (2020; Fig. 5) also show that Stockholm time series peaks at 2018 for 9 out of 10 of their analysed heat-wave indicators (Fig. 5 and 10 in Hoy et al. (2020))."

L201: What does a heatwave intensity of 65 K mean? - Meant is the index called HW95, analysed in the study by Hoy et al. 2020.
We write now: "An example is hot days (HD) with maximum temperature above 30° C which was observed with 8 days over the previous record (18 days) as well as the heat-wave duration with 22 days compared to 11 days previous record and a heat heat-wave intensity (HW_95, sum of daily excess maximum temperatures >P95) of 65 K compared to 35 K in 1975 (Hoy et al. 2020 Fig. 7). "

L206: The two opening clauses are repetitive. - We write now: "Zooming out to the larger scale by exploring the observed monthly data over Sweden, we note that the long-term MJJA 2018 average was 2.8 K above the 1981–2010 mean."

L288-289: The legend of Figure 3B seems to cover some of the data. - Thank you for pointing that out. It is true, the peak in the early 1860s reaches about 33 days and is covered by the legend box. We adjusted the position of the legend accordingly.

L391: Can these 1 in 20 and 1 in 100 statistics also be discussed in the results section? - The 1 in 20 and 1 in 100 are used here in the conclusions to summarize the presentation in the result section where we consistently use the 95th and 99th percentiles in the text. As we have chosen not to calculate exact return values for the various indices we have decided to keep these commonly used percentiles in the result section and consequently have not made any changes in the text.

L400: To occur in models? - We changed the sentence to: "Despite this, we still find such summers also to occur in models in the pre-industrial climate, although with a lower probability."

Figure S1: Models and observations appear to have different spatial coverage; are results affected by this? It tends to be standard in model intercomparisons to re-grid models to a common grid. - It is correct that the models and observations come with different spatial resolution.  As the re-gridding adds more uncertainties to the data and as we are calculating all analysis in this study on regional averages (Northern Sweden, Southern Sweden), we decided to not re-grid the data. By re-gridding data from a coarse grid to a finer grid we don't really add information. If we would upscale all data to the coarsest grid instead we would lose the details given by the higher resolved climate data.

Figure 2: The color differences corresponding to percentiles in the bars above the panels are difficult to see. Can you describe the kernel density fit used in more detail as well?  - We edit the caption which reads now:
"Probability distributions for monthly average temperature anomalies, calculated for 1861-1890 (dashed) and 1989-2018 (solid) against 1981-2010 for the Southern half of Sweden, for the months of May, June, July, and August, and seasonally pooled for JJA and MJJA. The bars in the upper part of each panel are a guide to compare the positions of percentiles for each ensemble and period. The opacity of the bars indicates in steps the .9, .95, .99, and 1 percentile ranges marked on the uppermost bar. Ensemble distributions are a kernel density fit, whereas the histogram for EOBS is based on actual data. The observed year 2018 for respective months/season is indicated by the vertical dotted black line."
For each ensemble we pooled together all ensemble members and calculated the kernel density estimate with a gaussian fit to gain a smooth curve.

**References**
Brunner, L., Schaller, N., Anstey, J., Sillmann, J., & Steiner, A. K. (2018). Dependence of present and future European temperature extremes on the location of atmospheric blocking. Geophysical Research Letters, 45, 6311– 6320. https://doi.org/10.1029/2018GL077837

Davini, P., and F. D'Andrea, 2016: Northern Hemisphere Atmospheric Blocking Representation in Global Climate Models: Twenty Years of Improvements?. J. Climate, 29, 8823–8840, https://doi.org/10.1175/JCLI-D-16-0242.1.

Dawson, A., & Palmer, T. N. (2015). Simulating weather regimes: Impact of model resolution and stochastic parameterization. Climate
Dynamics, 44. https://doi.org/10.1007/s00382-014-2238-x

Dunn-Sigouin, E., and S.-W. Son (2013), Northern Hemisphere blocking frequency and duration in the CMIP5 models, J. Geophys. Res. Atmos., 118, 1179–1188, doi:10.1002/jgrd.50143.

Hartung, K., Svensson, G., Kjellström, E. (2017) Resolution, physics and atmosphere–ocean interaction – How do they influence climate model representation of Euro-Atlantic atmospheric blocking?, Tellus A: Dynamic Meteorology and Oceanography, 69:1, DOI: 10.1080/16000870.2017.1406252

Jung, T.,Miller,M. J., Palmer, T. N., Towers, P.,Wedi,N. and co-authors. 2012. High-resolution global climate simulations with the ECMWF model in Project Athena: Experimental design, model climate, and seasonal forecast skill. J. Clim. 25(9), 3155–3172

Masato, G., B. J. Hoskins, and T. Woollings, 2013: Winter and Summer Northern Hemisphere Blocking in CMIP5 Models. J. Climate, 26, 7044–7059, https://doi.org/10.1175/JCLI-D-12-00466.1.

Meinshausen, M., Nicholls, Z., Lewis, J., Gidden, M. J., Vogel, E., Freund, M., Beyerle, U., Gessner, C., Nauels, A., Bauer, N., Canadell, J. G., Daniel, J. S., John, A., Krummel, P., Luderer, G., Meinshausen, N., Montzka, S. A., Rayner, P., Reimann, S., Smith, S. J., van den Berg, M., Velders, G. J. M., Vollmer, M., and Wang, H. J.: The SSP greenhouse gas concentrations and their extensions to 2500, Geosci. Model Dev. Discuss., https://doi.org/10.5194/gmd-2019-222, in review, 2019.

Moberg A, Bergström H, Ruiz Krigsman J, Svanered O. 2002: Daily air temperature and pressure series for Stockholm (1756-1998). Climatic Change 53: 171-212, doi:10.1023/A:1014966724670.

Müller, W. A., Jungclaus, J. H., Mauritsen, T., Baehr, J., Bittner, M., Budich, R., et al. (2018). A higher-resolution version of the Max Planck Institute Earth System Model (MPI-ESM1.2-HR). Journal of Advances in Modeling Earth Systems, 10, 1383– 1413. https://doi.org/10.1029/2017MS001217

Schaller N, Sillmann J, Anstey J, Fischer E M, Grams C M, and Russo S (2018) Influence of blocking on Northern European and Western Russian heatwaves in large climate model ensembles. Environ. Res. Lett. 13 054015, https://doi.org/10.1088/1748-9326/aaba55

Schiemann, R., and Coauthors, 2017: The Resolution Sensitivity of Northern Hemisphere Blocking in Four 25-km Atmospheric Global Circulation Models. J. Climate, 30, 337−358, https://doi.org/10.1175/JCLI-D-16-0100.1.

Sousa PM, Trigo RM, Barriopedro D, Soares PMM, Santos JA (2018) European temperature responses to blocking and ridge regional patterns. Clim Dyn 50:457-477.

Woollings, T., Barriopedro, D., Methven, J. et al. Blocking and its Response to Climate Change. Curr Clim Change Rep 4, 287−300 (2018). https://doi.org/10.1007/s40641-018-0108-z